# Efficient Fmoc-Protected Amino Ester Hydrolysis Using Green Calcium(II) Iodide as a Protective Agent

**DOI:** 10.3390/molecules27092788

**Published:** 2022-04-27

**Authors:** Renaud Binette, Michael Desgagné, Camille Theaud, Pierre-Luc Boudreault

**Affiliations:** Department of Pharmacology and Physiology, Faculty of Medicine and Health Sciences, Institut de Pharmacologie de Sherbrooke, Université de Sherbrooke, 3001 12e Avenue Nord, Sherbrooke, QC J1H 5N4, Canada; renaud.binette@usherbrooke.ca (R.B.); michael.desgagne@usherbrooke.ca (M.D.); camille.theaud@usherbrooke.ca (C.T.)

**Keywords:** green chemistry, ester hydrolysis, Fmoc, amino acid, solid-phase synthesis, medicinal chemistry

## Abstract

In order to modify amino acids, the C-terminus carboxylic acid usually needs to be protected, typically as a methyl ester. However, standard cleavage of methyl esters requires either highly basic or acidic conditions, which are not compatible with Fmoc or acid-labile protecting groups. This highlights the need for orthogonal conditions that permit selective deprotection of esters to create SPPS-ready amino acids. Herein, mild orthogonal ester hydrolysis conditions are systematically explored using calcium(II) iodide as a protective agent for the Fmoc protecting group and optimized for a broad scope of amino esters. Our optimized reaction improved on the already known trimethyltin hydroxide, as it produced better yields with greener, inexpensive chemicals and a less extensive energy expenditure.

## 1. Introduction

With the ever-growing popularity of solid-phase peptide synthesis (SPPS) using the Fmoc strategy, amino acid derivatization is a useful tool in structure–activity relationship (SAR) studies. It is usually the Boc-protected (backbone chain) amino acids that are used for the synthesis of those modified amino acids, but this strategy faces several drawbacks, as they do not absorb UV light, are sensitive in acidic environment, and must be deprotected and reprotected to Fmoc for SPPS while retaining other orthogonal protective groups [1]. Differently protected amino acids also exist (Alloc, Dde, Bn, etc.) but are usually more expensive and less available than Fmoc-protected ones and still face the same deprotection/reprotection conundrum.

Many reaction conditions involving the side chain are not directly compatible with the usual Fmoc-protected amino acids, as the free carboxylate can carry out nucleophilic side-reactions on alkyl halide, for example, if not adequately protected [2]. One way to circumvent this problem is to use the corresponding ester (typically the methyl ester). Since SPPS using the Fmoc strategy relies on the activation of the carboxylic acid, saponification is a mandatory step. Ester hydrolysis is a well-known reaction, but it is normally incompatible with either the Fmoc protecting group in basic conditions or the Boc group in an aqueous acidic environment. Here, we optimized an efficient ester hydrolysis method suitable for Fmoc-protected amino acid that incurs no epimerization.

We based our work on a previously reported article highlighting the use of calcium chloride as a protective agent of Fmoc-protected amino acids during hydrolysis of esters [3]. The original article did not, however, include a hypothesis on the mechanism of action, and there was no screening for the optimal solvent or saponification conditions. Herein, we expand on the hydrolysis conditions that seem not to affect base labile Fmoc protecting groups.

Previously reported saponification using trimethyltin hydroxide also proved to be successful at keeping Fmoc-protected amino acids intact while removing the ester [4]. This method, to our knowledge, is the gold-standard in ester hydrolysis of Fmoc-protected amino acids. Although this method is efficient for ester hydrolysis, it uses highly toxic and expensive organotin material that is known as a neurotoxin [5,6] and has a significant impact on mitochondrial respiration [7]. Previously reported trimethyltin hydroxide hydrolysis also uses 1,2-dichloroethane (DCE) as a solvent, which poses an ecological problem as chlorinated solvents are known for their carcinogenicity and accumulative properties [8]. Methods involving iodide salts (mostly magnesium) have also been reported for the hydrolysis of various esters [9] while preserving Fmoc groups [10]. The drawback of these methodologies is that they are incompatible with several other carbamates (Boc, Alloc) which are vital for SPPS.

Therefore, we optimized an ester hydrolysis method by systematically investigating solvents, bases, and salt additives that would also preserve Fmoc protecting groups. Our method not only uses greener non-chlorinated solvents but also completely removes the use of organotin compounds, heating, and tap water waste [11] and is more ecofriendly [12].

## 2. Material and Methods

All commercially available Fmoc-protected amino acids were purchased from Chem-Impex (Wood Dale, IL, USA), Matrix Innovation (Quebec, QC, Canada), and Combi-Blocks (San Diego, CA, USA) at the highest purity available. Other reagents and solvents were purchased from Matrix Innovation or Sigma–Aldrich (St. Louis, MO, USA) and were used as received.

### 2.1. Ester Synthesis

#### 2.1.1. General Esterification for Acid-Resistant Fmoc Amino Esters

The Fmoc-protected amino acid (0.3 mmol, 1 eq.) was added to MeOH, EtOH, or iPrOH (5 mL), and SOCl_2_ (0.36 mmol, 1.2 eq.) was added slowly at 0 °C. The reaction mixture was stirred at room temperature until completion (usually overnight), and the solution was concentrated under reduced pressure. The oil was then co-evaporated with MeOH three times.

#### 2.1.2. General Esterification for Acid-Sensitive Fmoc Amino Methyl Esters 

The Fmoc-protected amino acid (0.3 mmol, 1 eq.) was dissolved in dichloromethane (DCM) (5 mL) and *N*,*N*-diisopropylethylamine (DIPEA) (0.36 mmol, 1.2 eq.). MeI (0.36 mmol, 1.2 eq.) was added in one portion. The reaction mixture was stirred at room temperature until completion (usually five days). The solution was washed with 0.1 M aqueous HCl solution two times. The organic phase was then dried over Na_2_SO_4_, filtered, and concentrated under reduced pressure.

#### 2.1.3. Fmoc-Gly-OtBu Synthesis

Five hundred milligrams of glycine tert-butyl ester hydrochloride (3 mmol, 1 eq.) were dissolved in 5 mL of saturated aqueous NaHCO_3_, and a solution of 1.01 g of Fmoc *N*-hydroxysuccinimide ester (2.98 mmol, 1 eq.) in 5 mL of 1,4-dioxane was added dropwise. The resulting solution was stirred at room temperature overnight. Then, 1,4-dioxane was removed under vacuum. Fifteen milliliters of ethyl acetate were added, and the organic phase was washed with 0.1 M aqueous HCl solution two times, then dried over Na_2_SO_4_, filtered, and concentrated under reduced pressure to obtain 876 mg (83%) of a viscous liquid that crystalized spontaneously providing Fmoc-Gly-OtBu.

#### 2.1.4. Phth-Gly-OMe Synthesis

Three hundred and sixty-nine mg of methyl chloroacetate (3.4 mmol, 1 eq.) and 500 mg of phthalimide (3.4 mmol, 1 eq.) were dissolved in 5 mL of DMF; then, 650 μL of DIPEA (3.74 mmol, 1.1 eq.) was added. The solution was stirred at room temperature overnight and then air-dried overnight. The resulting slurry was dissolved in AcOEt and washed with 0.1 M aqueous HCl solution two times, saturated aqueous NaHCO_3_ two times, and brine two times; then, it was dried over Na_2_SO_4_, filtered, and concentrated under reduced pressure to give 650 mg (87%) of a white solid.

#### 2.1.5. Fmoc-Gly-OBn Synthesis

Five hundred mg of Fmoc-Gly-OH (1.68 mmol, 1 eq.) was dissolved in dry tetrahydrofuran (THF) (5.0 mL) in a round-bottomed flask thoroughly flushed with argon. Then, 312 μL of *N*,*N*′-dicyclohexylcarbodiimide (DCC) (2.0 mmol, 1.2 eq.) and 51 mg of 4-dimethylaminopyridine (DMAP) (0.42 mmol, 0.25 eq.) were added to the solution at 0 °C. Two hundred and ten mg of benzyl alcohol were slowly added, and the solution was stirred at room temperature overnight. THF was removed under vacuum. The resulting slurry was dissolved in ethyl acetate and washed with 0.1 M aqueous HCl solution two times; then, it was dried over Na_2_SO_4_, filtered, and concentrated under reduced pressure to give 551 mg (85%) of a white solid.

### 2.2. Calcium Iodide 5 M Solution Preparation

Calcium iodide (85 g, 30 eq. compared to the amino ester) was weighed and added portion-wise while stirring into 30 mL of cooled water using a salt/ice bath with frequent sonication to enhance solubility. The solubilization was exothermic. When the solution became oversaturated and formed a gel, small portions of the remaining volume of water were added until all the product was dissolved and the expected volume was attained. The resulting solution is unstable and should be used within a week.

### 2.3. Standard Conditions for Ester Hydrolysis

In a 24-well plate (model 931565-G-1X from Thomson Instrument, Oceanside, CA, USA), a magnetic stir bar and 100 µmol of amino ester (1 eq.) were added, followed by 600 μL of a 5 M solution of CaI_2_ (30 eq.), resulting in an insoluble mixture. Seventy-five microliters of a 2 M NaOH solution (1.5 eq.) were then quickly added, forming a white precipitate; then, 1.6 mL of organic solvent was subsequently added (1.6 mL/100 μmol amino ester), resulting in a 2.3:1 organic solvent for aqueous mixture. The 24-well plate was covered using adhesive aluminum foil to avoid evaporation and left to stir for the desired amount of time.

### 2.4. Reaction Quench for Sampling

Twenty microliters of the crude reaction mixture were added to 200 µL of quench solution (HCl 0.01 M in MeOH) to bring the pH down to 3–4. Solutions were found to be stable using this quench mixture (Appendix A). The quenched mixture was subsequently assayed for composition by UPLC-MS.

### 2.5. UPLC-MS Analysis of Crude Material

Previously quenched solutions were analyzed using a Waters UPLC-MS system (column Acquity UPLC^®^ CSHTM C18 (2.1 × 50 mm) (Agilent Technologies, Santa Clara, CA, USA) packed with 1.7 μm particles) using ACN and water + 0.1% formic acid. The method used was 0 to 0.2 min: 5% ACN; 0.2 to 1.5 min: 5% to 95% ACN; 1.5 to 1.8 min: 95% ACN; 1.8 to 2.0 min: 95% to 5% ACN; 2.0 to 2.5 min: 5% CAN in the ESI+ mode. Percent yields were assayed using integration in the MassLynx software (Agilent Technologies, Santa Clara, CA, USA). Integration of the peaks was carried out by considering all UV-active peaks (except for the solvent front). The results (%) for acid, ester, and dibenzofulvene might not add up to 100% if a significant side-reaction occurred. All data was analyzed using GraphPad Prism version 9.3.1 (San Diego, CA, USA).

### 2.6. NMR Data Acquisition

Nuclear magnetic resonance (NMR) spectra were obtained using a Bruker Ascend 400 spectrometer (400 MHz) (Billerica, MA, USA). Chemical shifts are given in parts per million (ppm) (δ relative to the residual solvent peaks for both 1H and 13C). All NMR data were collected using DMSO-d6 as a solvent, and residual solvent signal (DMSO) were used as an internal reference for 1H (δ = 2.5 ppm) and 13C (δ = 39.5 ppm) spectra.

### 2.7. Extraction of the Final Product

The reaction mixture was neutralized using a 2 M aqueous HCl solution until a neutral to slightly acidic pH was obtained, and it was subsequently concentrated under vacuum. Aqueous HCl 0.01 M was added and extracted twice using AcOEt. The organic fractions were combined and washed with saturated Na_2_S_2_O_4_ and brine and subsequently dried over Na_2_SO_4_. The organic phase was filtered and evaporated in vacuo to give the Fmoc-protected amino acid in a quantitative recovery.

### 2.8. Chiral HPLC Methodology

The previously extracted material and the pure amino acids (L, D, and racemic) were solubilized in EtOH with 0.1% of formic acid at a concentration of 2 mg/mL. The resulting solution was analyzed using a Shimadzu Prominence 20A (Kyoto, Japan) (column Chiralpak AD-H (250 × 4.6 mm) packed with 5 µm particles) with a SPD-M20A UV detector using 10% iPrOH + 1% formic acid/hexanes as the mobile phase at 1 mL/min. All results were analyzed using a Shimadzu LC solution (Kyoto, Japan) and integrated using 254 nm UV spectra. Integration of the compound was compared to the pure amino acid standards.

## 3. Results and Discussion

### 3.1. Nature of the Organic Solvent

The organic solvent choice was first investigated, as the solubility of the salt was suspected to be an important factor for Fmoc preservation (Figure 1). Solvent selection was carried out solely using water-soluble solvents, as a biphasic medium would compromise the reaction. Previously published conditions call for 19 eq. of CaCl_2_ in iPrOH, hence, the number of equivalents of calcium salt used in this experiment. These conditions showed deprotection of the Fmoc and an important side-reaction that was later identified as the formation of Fmoc-Gly-OiPr, as it showed the same retention time and mass via UPLC-MS. Saponifications in basic conditions are known to be in equilibrium with transesterification [13], which means that the isopropyl ester, being more sterically hindered than the methyl ester, slows down its hydrolysis, resulting in the non-negligible appearance of this compound after 4 h.

After 4 h of reaction, four solvent conditions showed less than 10% deprotection of the Fmoc: acetone, THF, dioxane, and iPrOH. THF and dioxane, however, offered little hydrolysis product (<15%) to the corresponding carboxylic acid and were therefore discarded as potential candidates for solvent selection. Yet, acetone seemed to yield similar hydrolysis results compared to the previously reported iPrOH, with no formation of the isopropyl ester intermediate, which was a clear positive. Isopropanol in basic solution can lead to transesterification, which creates a more hindered, less reactive molecule. The isopropyl ester therefore accumulates in the mixture, leading to smaller yields.

It must be noted that DMSO, ACN, MeOH, and DMF showed little to no precipitation after 4 h and displayed the highest amount of Fmoc deprotection. The high solubility of Ca(OH)_2_ is suspected to be the cause of this undesired reactivity and is discussed below. Acetone is also considered to be a recommended or problematic solvent for green chemistry [14] and ranks at approximately three using the EHS method [15].

Following this experiment, acetone was identified as the most promising organic solvent for the following experiments, as it showed better hydrolysis, lower deprotection of the Fmoc group, and a lower ecological footprint compared to the other tested organic solvents.

### 3.2. Nature and Concentration of the Inorganic Base

The nature of the base was also thought to be an important factor in the saponification and deprotection of the Fmoc group, as lithium has been reported to be important for ester hydrolysis in aqueous conditions [16]. Fmoc-Gly-OMe hydrolysis was further investigated by using increasing concentrations of inorganic hydroxides, namely, LiOH, NaOH, and KOH (Figure 2).

The three tested hydroxides showed a similar pattern of deprotection and hydrolysis after 4 h. Fmoc deprotection was observed for all concentrations in a dose-dependent manner, and ester hydrolysis was observed as maximal using at most 2 eq. of base (Appendix A). NaOH showed marginally more saponified product and less dibenzofulvene production. Although the yields were similar, 1.5 eq. of NaOH was selected for the following assays because it is less expensive and more commonly distributed amongst laboratories while retaining the minimal amount of dibenzofulvene formation. LiOH and KOH, however, are probably equivalent towards ester hydrolysis in our conditions.

### 3.3. Initial Considerations of Ca(OH)_2_

One of the initial hypotheses for Fmoc protection was that the calcium salt acted as a hydroxide trap and precipitated, reducing the quantity of hydroxides in solution and, therefore, its basicity. As Ca(OH)_2_ was identified as a potential source of slightly soluble hydroxides and as a possible explanation for the behavior observed in previously described literature [3], ester hydrolysis was tested using calcium hydroxide in a dose-dependent manner.

Ca(OH)_2_ was not able to completely hydrolyze the amino ester with less than two equivalents or four equivalents of hydroxides (Figure 3). Dibenzofulvene also appeared significantly less compared to using solely NaOH (Appendix A). Calcium chloride also seemed to have a negative effect on Fmoc protection when used in conjunction with calcium hydroxide, as its use showed a more significant amount of dibenzofulvene in the reaction mixture (Appendix A). This suggests that although calcium hydroxide seems promising as the main explanation for Fmoc preservation, this alone does not explain the whole phenomenon. Further research must be conducted on the use of calcium hydroxide as a mild reagent for ester saponification.

### 3.4. Nature of the Salt

#### 3.4.1. Nature of the Cation Using Salt Chlorides

Several other cation chlorides were investigated, since cation hydroxide formation was thought to be of importance as shown by experiments involving calcium hydroxide (Figure 3). In addition, the Lewis acid potential might have an impact on hydrolysis kinetics [17]. Therefore, several compounds were investigated for their potential Fmoc protective characteristics and saponification. All chloride salts were employed at approximately the highest concentrations possible using our previously described method, as the impact of the salt seemed larger at higher concentrations according to previously described literature [3].

None of the chloride salts were able to reproduce the results, performing similarly to calcium chloride. Several salts greatly hindered saponification rates (i.e., MgCl_2_, LaCl_3_, ZnCl_2_, FeCl_2_, and FeCl_3_), as they are suspected to react with sodium hydroxide to form their own metal hydroxides, which are almost insoluble in water. This observation confirms that reactions involving added calcium salts with solubilized hydroxide to form calcium hydroxide are, indeed, possible and might be part of the answer for Fmoc preservation (Figure 3). Monovalent cations, which were not suspected to act as protective agents for Fmoc (LiCl, KCl), showed high deprotection rates and high hydrolysis yields. Those yields were similar to NaOH alone in solution (Figure 2). Finally, BaCl_2_ showed high conversion, but more Fmoc deprotection than CaCl_2_. Barium hydroxide was the most soluble hydroxide counterpart of the multivalent cations (Table 1) and was the only one able to completely hydrolyze the starting material. Barium hydroxide has already been shown to be a saponification agent [18] and is suspected to play a similar role to calcium hydroxide in our experiments. Although barium has a positive impact on ester hydrolysis, calcium showed approximately half the Fmoc deprotection. Calcium was, therefore, considered as the better protective agent for Fmoc groups in basic ester hydrolysis. The reactivity of the cation chloride and the solubility of the cation hydroxide are suggested to be of importance for Fmoc preservation and ester hydrolysis.

#### 3.4.2. Nature and Concentration of the Anion Using Calcium Salts

Using previously identified calcium as the better cation, we screened the quantity of calcium salts that would be optimal for ester saponification and protection of the Fmoc. The nature of the calcium salt was also investigated, using corresponding halogen derivatives.

CaF_2_ assisted hydrolysis (Figure 4A) showed little to no improvement when compared to the basic hydrolysis conditions with no calcium salt (Appendix A). Moreover, it must be noted that CaF_2_ showed very poor solubility in aqueous solution, leading to UPLC-MS problems. This explains the limited number of equivalents screened. Previously reported CaCl_2_ showed a complex polynomial curve (Figure 4B) that did not fit with the predicted pattern of hydrolysis and deprotection. This result was repeated twice, with the same pattern appearing every time. Solubility problems at higher concentrations are suspected to be the main reason for that reactivity shift. Interestingly, CaBr_2_ and CaI_2_ (Figure 4C,D) show a similar pattern of deprotection and hydrolysis, having enhanced protection of the Fmoc compared to the reference CaCl_2_ (Figure 4B), especially at 19.2 eq., which our group identified as a problematic region of the graph (identified by the vertical, dotted line). Better solubility in acetone and faster dissociation are suspected to be the main reasons why they react more predictably than CaCl_2_ and CaF_2_ at higher concentrations. CaI_2_ also showed slightly less deprotection compared to CaBr_2_. Thirty equivalents of calcium salt were also suspected to yield more reliable results, as Pascal et al. [3] showed a link between salt concentration and hydrolysis yields. Ten equivalents of CaCl_2_ also proved to create suitable hydrolysis conditions according to Figure 4B; however, the results were quite unstable around that concentration. In our case, only a small window of 4 h was observed, but we believe that during shorter or longer reaction times, 30 equivalents would create better hydrolysis conditions. Considering these results, the rest of the study was carried out with 30 eq. of CaI_2_, as it showed high saponification rates and the lowest rates of deprotection after 4 h.

### 3.5. Reaction Kinetics

With both the optimal quantity of base and salt identified, a kinetic study was performed to determine the optimized reaction times using our acetone/water mixture and 30 eq. of CaI_2_ compared to previously published literature.

When standard basic hydrolysis conditions were compared to reactions kinetics involving calcium salts (Figure 5), an increase in the carboxylic acid yield can be observed. This result is consistent with previous assays, as the presence of calcium chloride is shown to reduce the quantity of the final product by deprotecting it, especially at the concentration of 19 eq. (Figure 4B). However, using our newly optimized conditions lead to a final yield of 81% at 24 h of reaction time. This reaction time was chosen as it showed better conversion to the corresponding carboxylic acid (data not shown on graph). This result was compared to trimethyltin hydroxide (Appendix A) and showed an enhanced yield of approximately 20% while being significantly less toxic [5,6,7,8].

### 3.6. Scope of the Reaction

Before applying this optimized methodology to most standard amino acids, we investigated the potential racemization of one amino acid, as amino esters are known to epimerize under alkaline conditions [19]. Fmoc-Phe was selected for this assay, as it bears an acidic alpha proton, determined by its high exchange rate [20], is readily separated using Chiralpak AD columns in HPLC experiments [21], and is UV active in its deprotected form. Chiral HPLC experiments showed no epimerization after being subjected to our optimized conditions (Appendix A).

To highlight the potential of the hydrolysis of esters in the presence of Fmoc, we applied our optimized conditions to a variety of amino acids. Several of them with less common protective groups were also considered, especially alloc and allyl esters, which are important orthogonal protection strategies, for instance, in macrocyclization [22]. Fmoc-Aib-OMe and Fmoc-Gly-OEt, -OiPr, -OtBu, and -OBn were also subjected to this improved hydrolysis condition to evaluate the impact of steric hindrance on reaction completion, as most hindered esters hydrolyze more slowly [23]. Finally, Phth-Gly-OMe was also tested to observe the impact of CaI_2_ on phthalimide hydrolysis.

Our study showed high hydrolysis rates for all standard amino esters (Table 2, category 1) with an average of 84%. Our reaction conditions seemed compatible with usual side-chain protective groups (i.e., Pbf, Boc, Trt, tBu, and Alloc). Lower hydrolysis rates were also observed with β-hindered amino esters, namely, Ile, Thr, and Val (Table 2, category 2). Reaction conditions seemed to be sensitive to steric hindrance, showing an average yield of 52.3% for these three amino acids only. It should be noted that Fmoc deprotection did not significantly appear more than category 1 amino acids, meaning that the lower conversion was due to the starting material that had not yet been consumed. Longer reaction times could yield better results for more hindered amino acids.

Different esters were also assessed, leading to variable results. Interestingly, -OEt, -OiPr, and -OBn esters were readily hydrolyzed by our method, showing a low impact of steric hindrance (Table 2, category 3). The highly hindered Fmoc-Gly-OtBu ester showed significantly lower conversions compared to other esters. Finally, as suspected, Fmoc-Aib-OMe showed slower hydrolysis compared to Fmoc-Gly-OMe in similar conditions for a final yield of 32.9%. Quaternary carbons, especially in Aib, are known to slow nucleophilic reactions with the terminal amine [24] as well as the terminal carboxylic acid [25].

Allyl esters and phtalimide protecting groups were readily hydrolyzed by our method, showing incompatibility with these protecting groups (Table 2, category 4). Standard deprotection conditions for Phth used in the literature involve either hydrazine, a highly nucleophilic diamine, or basic conditions using low concentrations of sodium hydroxide [26]. Reducing conditions using sodium borohydride have also been reported [27]. All those methods are also expected to affect several SPPS protecting groups.

This method was used for scale-up purposes on 3 g of Fmoc-Gly-OMe. Addition of acetone heated up the reaction mixture significantly, which forced early completion, showing complete disappearance of the starting material within 1 h. This scale-up provided 2.63 g of final product (92% yield) with a final UV purity of 95.6% using only extractions.

### 3.7. Example of Newly Accessible Amino Acid Chemistry

Side chains containing amines or anilines must be Boc-protected to limit protecting group manipulation during SPPS. Previously described SPPS-ready amino acids [28] were synthesized using aziridine building blocks on solid phase. This synthesis method requires extensive synthetic steps and is not atom efficient, as solid-phase trityl bound to polystyrene is used. Nonetheless, this method showed an average yield of ~60% for the tested N^β^-alkyl substituents. Using calcium iodide as an additive in ester hydrolysis permits saponification of compound containing both Fmoc and Boc protecting groups simultaneously, as seen using Fmoc-Lys(Boc)-OMe (Table 2) in acceptable yields. The result is a more efficient synthesis for theoretical custom amino acids containing amines on the side chain (Figure 1).

## 4. Conclusions

In conclusion, our newly optimized method differs from previously reported orthogonal ester hydrolysis by its use of acetone, sodium hydroxide, and calcium iodide as saponification conditions. This method permits high yields of hydrolysis without significant cleavage of Fmoc and is orthogonal to most standard protecting groups while retaining stereochemistry. This method is suspected to involve Ca(OH)_2_ as the reactive intermediate, which acts as a slightly soluble hydroxide trap, lowering the concentration of hydroxides in solution. Our methodology also improves on the trimethyltin hydroxide hydrolysis, which uses toxic and expensive organotin material as well as bio-accumulative chlorinated solvents. The use of calcium iodide and sodium hydroxide in conjunction yields better acid conversion. This method allows for a straightforward chemistry for synthetic amino acids containing amines on the side chain, as Fmoc and Boc protecting groups are orthogonal to our saponification conditions. We believe that this optimized hydrolysis method may be a useful tool for medicinal chemists to further simplify amino acid modifications.

## Data Availability

Data is contained within the article or Appendix A.

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
