# Peer review of "Efficient Fmoc-Protected Amino Ester Hydrolysis Using Green Calcium(II) Iodide as a Protective Agent"

_molecules, 2022, doi:10.3390/molecules27092788_

Round 1

Reviewer 1 Report

The authors present an extensive and systematic study on ester hydrolysis of Fmoc-protected amino acids, involving screening of solvent, base and calcium salts as protecting additives.

This work is of great interest to those involved in peptide synthesis research as it addresses the very common need for orthogonal deprotection, especially when dealing with sequences involving multiple protecting groups of different sensitivity.

The data is clearly presented and the findings in this report constitute an alternative and efficient strategy for ester hydrolysis with minor Fmoc cleavage side reaction. Therefore, publication is recommended.

Minor correction required: in page 7, line 2, reference to Fig S2 is incorrect.

Author Response

Referenced correctly the figure

Reviewer 2 Report

It is a well structured manuscript, which reports original data; the text is well followed.

1) In my opinion, this research is relevant and interesting. 2) I considered the topic of the research, as well as its scientific soundness and interest to the readers as AVERAGE, in comparison to other published works. 3) I think that the conclusions, of this research are consistent with the evidence  and the presented arguments. 4) However, do they address the main question posed, more or less satisfactorily. 5)  In my opinion, the key highlights within the manuscript that are worth of mention in the field are:      (a) The need for orthogonal conditions which will permit selective deprotection of esters to create            Solid-Phase-Peptide-Synthesis-ready amino acids,     (b) Under reaction conditions which are compatible with Fmoc or acid-labile protecting groups, and     (c) The proposed reaction optimization permits improved yields, using both green and cheaper chemicals,           along with reduced energy demands. 

Author Response

Thank you for your input

Reviewer 3 Report

The manuscript by Binette et al. describes a procedure for preparing deprotected C-terminal amino acids by hydrolyzing an ester in the presence of the labile Fmoc group. Although the novelty is perhaps a little low (as there are examples of calcium salts being used for this sort of hydrolysis before), this manuscript should be published in Molecules, as its results are of potential high utility to the peptide chemistry and synthetic chemistry community. I do think that some changes should be made before publication, particularly, some examples on a scale larger than micromolar should be provided as proof that the reaction can be used for preparative synthesis.

Comments:

  1. It would be easier to follow the manuscript if all compounds in the manuscript and the supporting info had a number associated with them (e.g., 1a, 1b, etc.), instead of having to match the chemical names in the manuscript to those in the supporting information, when a lot of the chemical names and abbreviations are similar.
  2. Although the methodology has been thoroughly investigated and works well, it would be pertinent to see this applied on a larger scale (ca. 1 g) for several examples, to see how the yields scale up and how the reaction works (is it slower? It is it exothermic?), as someone will inevitably want to use this method for preparing a large amount of an amino acid. These are useful precursors for drug or natural product synthesis; please include these examples.
  3. On what scale was the “custom amino acid” synthesized? It mentions “average yield” – average of how many replicates? Is it highly variable? These questions should be answered in the text.
  4. In the section “UPLC-MS analysis of crude material”, please state the mass spectrometer ionization source and in which mode (+/-) data was acquired. Also, please state the NMR, field strength, and NMR solvents used in the experiments.
  5. Please add “Aib” to the list of abbreviations, as this does not appear to be a common abbreviation.
  6. Supporting info figure S4 is not mentioned in the manuscript text, and should be.
  7. Figure 6 should be “Scheme 6”.
  8. Keep your citation style consistent and uniform. Sometimes full author names are given, sometimes just initials.

Author Response

Thank you for your input

  1. We think having the compounds named is easier, as it speaks of itself, without the need for referencing in the text. Several compounds have been tested and amino acid notation is well known to peptide and amino acid chemists. We believe that using references forces readers to read both the table and the text at the same time which makes for confusing reading
  2. That comment was very useful, as scaleup was not investigated. We therefore synthezied 3g of Fmoc-Gly-OMe and investigated the scale-up purposes. Added a paragraph on p.5, lines 30-36 for CaI2 solution preparation, one row in table 2 for scale-up results and a paragraph on p. 17, lines 26-30 for explanation of the scale-up.
  3. This custom amino acid was not synthesized, only theorically accessible now using our chemistry. Fixed the scheme by using dashed arrows instead of full arrows.

  4. Added a paragraph in materials and methods (2.5 NMR data acquisition). Also added ESI+ to UPLC-MS characterization.

  5. Added Aib to the abbreviation list

  6. Added a small paragraph on p15, lines 6-7 which mention figure S4

  7. Changed figure for scheme

  8. Removed all first names

Reviewer 4 Report

This manuscript described novel ester hydrolysis method for Fmoc-protected amino acids. The author revealed that calcium(II) iodide efficiently preserved Fmoc protecting group during a base hydrolysis reaction, and that the method using calcium(II) iodide preformed high yields of hydrolysis without deprotection of most standard protecting groups for SPPS as well as Fmoc group and racemization. I think the results will attract the attention of the journal readers. But I feel description of reaction conditions is insufficient. The paper is worthy of publication on Molecules after major revision.

Major comments:

1) How many times did the author perform each experiment? I think the author should perform it more than three times to indicate the probability of the result.

Minor comments:

1) Figure S1: Details of ester hydrolysis reaction were unidentified. There is no information about additive and organic solvent used in the reaction. I think the author checked stability of a hydrolysis reaction mixture in the quench solution in this figure. Is it correct? If correct, why did not hydrolysis occur?

2) page 6, 2.6, line 6, “254 nM”: Change “nM” to “nm”.

3) page 7, line 2: Figure S2 in the supplementary information does not show the formation of Fmoc-Gly-OiPr. I think the author failed to attach the data about Fmoc-Gly-OiPr. Please confirm it.

4) Figure 2: Details of the reaction were unidentified. There is no information about additive and solvent used in the reaction. Was CaCl2 added in the reaction mixture? The author should describe the details of the reaction.

5) page 8, line 5 from the bottom, “at least 2 eq.”: I think that maximal rate of hydrolysis was observed using at most 2 eq. of base. Is it wrong?

6) Figure S2: The horizontal axis should be drawn from 0 on the vertical axis.

7) Figure 3: Details of the reaction were unidentified. There is no information about substrate and solvent used in the reaction. The author should describe the details of the reaction.

8) Table 1: Details of the reaction were unidentified. There is no information about solvent used in the reaction. The author should describe the details of the reaction.

9) page 10, line 2 from the bottom, “Fig. 4”: Figure 4 does not indicate importance of calcium hydroxide. I think the author miswrote the reference. Please confirm it.

10) Figure 4: Details of the reaction were unidentified. There is no information about solvent used in the reaction. The author should describe the details of the reaction.

11) page 12, line 2, “Fig. S2”: Figure S2 does not indicate CaF2 effect. I think that the author miswrote the reference, and that correct reference is Fig. 4A. Please confirm it.

12) page 12, line 10, “19.2 eq.”: I think the best result was observed using 10 eq. CaCl2 in Figure 4B and the result was better than that using high concentration CaBr2 and CaI2. Why did the author compare the results among calcium salt at 19.2 eq.? The author should describe the result using 10 eq. CaCl2.

13) page 12-14, 3.5: The author does not discuss the optimal reaction time. The author should describe a conclusion in this section.

14) Table 4: Table 2 and 3 don’t exist in this paper. Correct the table number.

15) Table 4: Why was the reaction time 24 h? I think the best result was observed after about 2 h in Figure 5C. The author should explain the reaction time.

16) Table 4: Why did the author use 30 eq. CaI2? I think lower concentration CaI2 also can show high saponification rates and the lowest rates of deprotection. The author should explain the equivalent amount of CaI2 used in the reaction.

17) Table 4: The author should add the data of “%dibenzofulvene” in Table 4.

18) page 15, line 3 from the bottom, “category 1”: The label “category 1” doesn’t exist in Table 4. The author should add “category 1” in Table 4 or change “category 1” in the text to “standard amino methyl esters”. The same applies to the following “category 2” and “category 3”.

19) page 16, line 3-4, “Fmoc deprotection did not significantly appear”: The author should describe specific amount of “%dibenzofulvene”. See comment 17.

20) page 16, 3rd paragraph: What is the rates of deprotection of allyl ester and phthalimide? The author should describe specific amount.

21) Figure 6: The author should add the yields of these compounds. Readers cannot assess whether the method is useful or not as it is.

22) Figure 6: There is no compound data of these compound and reaction conditions in the text. Are all these compounds known? At least, the author should describe the reaction conditions in section 2.

23) page 20, reference 27, line 3, “no 42”: “o” is superscripted. Remove superscript. The same applies to the following reference 28.

24) Figure S4: Figure S4 is not referenced in the main text. Is this figure necessary?

Author Response

Thank you for your generous input on our article

  1. Most experiments were done once, as it showed very reproducible results, as confirmed by timecourses which fit clearly into square root type regression (r2 = 0.88 in most cases for apparition of the carboxylic acid and disappearance of ester). Some results (Fig.4B, 19 eq of CaCl2) were made several times as they seemed abnormal, and showed similar results without fail, we therefore added a triplicate on that point alone, which was problematic.
  2. We took a reaction mixture and immediately quenched it using 1 eq. HCl (compared to NaOH), hence the lack of hydrolysis. We wanted to investigate the stability of the neutralized solution for time constraints reasons. Since the quantity of compounds did not evolve over time post-neutralization, we were not constrained to inject via UPLC quickly.

  3. Changed nM to nm

  4. Changed the paragraph so it would remove that figure from supplementaries. Instead proved the presence of Fmoc-Gly-OiPr using MS and LC retention time.

  5. Described figure 2 more adequately

  6. Changed “at least” for “at most”

  7. Changed the graph to fit 0 on the horizontal axis

  8. More detailed hydrolysis procedure was written in materials and methods (2.2). Figure 3 description was also corrected
  9.  Standard hydrolysis conditions are details in material and methods, section 2.2. We changed the title of that section so it is less confusing.

  10. Changed Fig.4 for Fig.3
  11. Reference was correctly written, but sentence was confusing, changed it so it's less confusing

  12. 19.2 eq. was the amount used by Pascal et al. which was shown as problematic. We agree that 10 eq. of CaCl2 shows great results, but is quite unstable, as small variations of the quantity can greatly decrease results. CaI2 shows stability of results, which was more suitable for our experiments.

  13. Added a small paragraph on page 15, lines 6-7 for choosing T = 24 hours for the final screen on all amino acids

  14. Table number corrected

  15.  The 24h time point was not shown on Fig. 5 as it added no visual benefit to the graph but shows higher completion for 24h. Although dibenzofulvene appeared more significantly, ester disappearance was complete. Added a small paragraph on page 15, line 6-7

  16. Added a paragraph at page 13, lines 6-8 which explains 30 equivalents.

  17.  Data added to table 2
  18.  Added numeral labels for the entries in the “category” column

  19. Wrote a small sentence that fixes that, p. 17, line 8-11
  20.  Phth-Gly-OH was observed only in traces in the reaction mixture using MS identification and showed complex reaction mixture. I think readers of our article only want to know if  our method is suitable for Allyl esters and Phth protection, which it is not. 

  21. This compound has never been made, only proposed as new attainable chemistry using our method. Reaction arrows were changed from full arrow to dashed arrows to show that it has not been done

  22. This compound has never been made, only proposed as new attainable chemistry using our method. Reaction arrows were changed from full arrow to dashed arrows to show that it has not been done

  23. Removed superscript

  24. Added a small paragraph on p15, lines 6-7 which mention figure S4

Round 2

Reviewer 4 Report

The manuscript has been revised well, but I think it needs a little more revision. As I think results of this paper will attract the attention of the journal readers, the paper is worthy of publication on Molecules after minor revision.

Minor comments:

1) Table 1, and author response 9: Solvent used in the reaction was not described in the legend of Table 1 and method 2.3. The author should specify the solvent.

2) Author response 12: I can understand author’s explanation and I think the author should describe the discussion in main text.

3) Author response 13 and 15: I think it is inappropriate that the data (% acid, % ester and % dibenzofulvene) after 24 h, which is the most important result, is not shown in the paper. I agree author’s opinion that there is no visual benefit to add the 24 h time point in Figure 5, but the author should show the data after 24 h, which can be compared with the date at shorter time, in a different way.

4) page 16, line 26, “3g”: Insert a space after “3”.

5) Scheme 6 and author response 21: I think it is inappropriate that the imaginary scheme is shown in the main text, even if dashed arrows are used in the figure. The author should not show Figure 6. If the author insists showing the scheme, the author should clearly describe that these compounds are not synthesized in this study.

Author Response

  1. Added solvent conditions in description of table 1 and methods 2.3
  2. Added description p. 13, lines 4-6 to talk about CaCl2 10eq.<
  3. We agree on this comment and created the figures with a break after 4 hours to show T = 24h. The graph still looks good and now shows relevant data.
  4. Space added
  5. Added "theoretical" in both the text and description of scheme 6